# Self-Supervised Deep Learning on Point Clouds by Reconstructing Space

**Jonathan Sauder**
Hasso Plattner Institute
Potsdam, Germany
jonathan.sauder@student.hpi.de

**Bjarne Sievers**
Hasso Plattner Institute
Potsdam, Germany
bjarne.sievers@student.hpi.de

## Abstract

Point clouds provide a flexible and natural representation usable in countless applications such as robotics or self-driving cars. Recently, deep neural networks operating on raw point cloud data have shown promising results on supervised learning tasks such as object classification and semantic segmentation. While massive point cloud datasets can be captured using modern scanning technology, manually labelling such large 3D point clouds for supervised learning tasks is a cumbersome process. This necessitates methods that can learn from unlabelled data to significantly reduce the number of annotated samples needed in supervised learning. We propose a self-supervised learning task for deep learning on raw point cloud data in which a neural network is trained to reconstruct point clouds whose parts have been randomly rearranged. While solving this task, representations that capture semantic properties of the point cloud are learned. Our method is agnostic of network architecture and outperforms current unsupervised learning approaches in downstream object classification tasks. We show experimentally, that pre-training with our method before supervised training improves the performance of state-of-the-art models and significantly improves sample efficiency.

## 1 Introduction

Point clouds provide a natural and flexible representation of objects in metric spaces. They can also be easily captured by modern scanning devices and techniques. Algorithms that can recognize objects in point clouds are crucial to countless applications such as robotics and self-driving cars. Traditionally, systems for such tasks have relied on the approximate computation of geometric features such as faces, edges or corners [31, 11] and hand-crafted features encoding statistical properties [3, 27]. However, these approaches are often tailored to specific tasks, thus not providing the necessary flexibility for modern applications. Recently, Convolutional Neural Networks (CNNs) which are domain-independent have shown promising performance on point clouds in supervised learning tasks such as object classification and semantic segmentation, outperforming conventional approaches [23, 24, 33, 16].

The advent of scalable 3D point cloud scanning technologies such as LiDAR scanners and stereo cameras gives rise to massive point cloud datasets, possibly spanning large entities such as entire cities or regions. However, manually annotating such massive amounts of data for supervised learning tasks such as semantic segmentation poses problems due to typical real-world point clouds reaching billions of points and petabytes of data, opposing the innate limitations of user-interfaces for 3D data labelling (e.g. drawing bounding boxes) on 2D screens. Therefore, it is of large interest to develop methods which can reduce the number of annotated samples required for strong performance on supervised learning tasks.

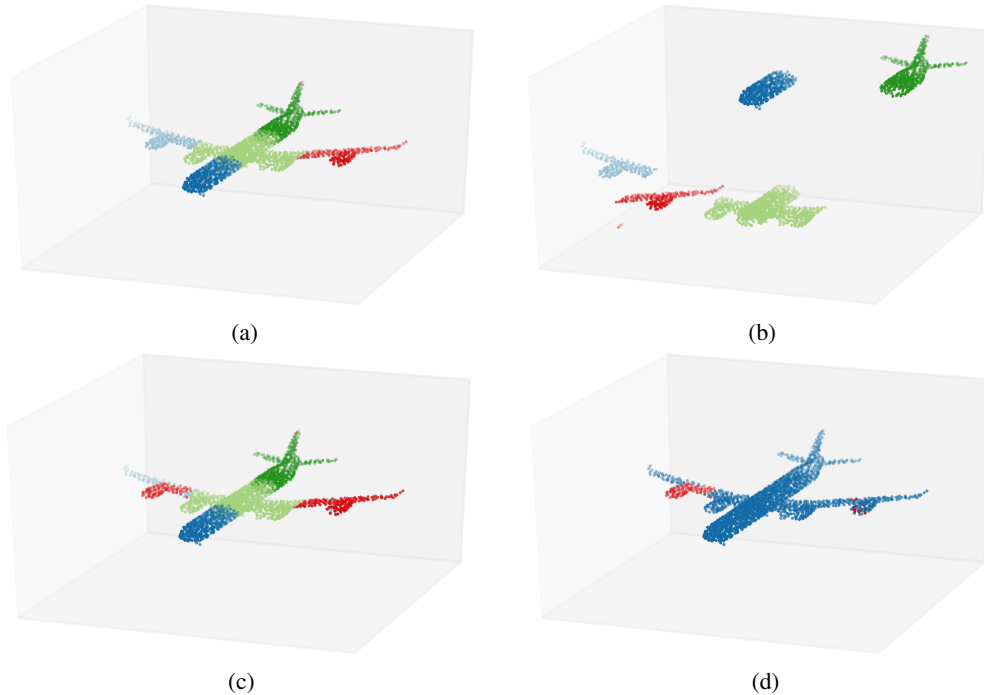

Figure 1: A visual example of the proposed self-supervised learning task. (a) The original object is split into voxels along the axes, each point is assigned a voxel label. (b) The voxels are randomly rearranged. (c) A neural network predicts the voxel labels, here visualized with the original point positions. (d) Points with correctly predicted voxel labels (blue) and misclassifications (red).

Unsupervised or self-supervised learning approaches for deep learning have shown to be effective in this scenario in various domains [10, 20, 13, 7, 21, 9]. On point clouds, self-supervised approaches have been largely focused on applying either Autoencoders [13] or Generative Adversarial Networks (GANs) [10]. While GAN-based approaches have not been successfully applied to raw point cloud data due to the non-triviality of sampling unordered sets with neural networks, Autoencoders for point clouds rely on possibly problematic similarity metrics [1].

In this work we address these limitations and present a self-supervised learning method for neural networks operating on raw point cloud data in which a neural network is trained to correctly reconstruct point clouds whose parts have been randomly displaced. An example of the proposed self-supervised task given in Figure 1. The proposed method is agnostic of the specific network architecture and can be flexibly used to pre-train any deep learning model operating on raw point clouds for other tasks. In a series of experiments, we show that powerful representations of point clouds are obtained from self-supervised training with our method. Our method outperforms previous unsupervised methods in a downstream object classification task in a transfer learning setting. We also explore per-point features and show pre-training with our method improves the performance and sample efficiency in supervised tasks. To highlight our main contributions:

- We present an architecture-agnostic self-supervised learning method operating on raw point clouds in which a neural network is trained to reconstruct a point cloud whose parts have been randomly displaced. Our method avoids computationally expensive and possibly flawed reconstruction losses or similarity metrics on point clouds.

- We demonstrate the effectiveness of the learned representations: our method outperforms state-of-the-art unsupervised methods in a downstream object classification task. Pre-training with our method improves results in all evaluated supervised tasks.

## 2  Related Work

### 2.1  Deep Learning on Point Clouds

Deep neural networks have shown impressive performance on regularly structured data representations such as images and time series. However point clouds are unordered sets of vectors, therefore exemplifying a class of problems posing challenges for deep learning for which the term *geometric deep learning* [4] has been coined. Although deep learning methods for unordered sets [32, 39] have been proposed and also applied to point clouds [25], these approaches do not leverage spatial structure.

To address this problem, popular point cloud representations suitable for deep learning include volumetric approaches, in which the containing space is voxelized to be suitable for 3D CNNs [18, 22, 36], and multi-view approaches [28, 30], in which 3D point clouds are rendered into 2D images fed into 2D CNNs. However, voxelized representations can be difficult to use when the point cloud density varies, and as such are constrained by the resolution and limited by the computational cost of 3D convolutions. Despite multi-view approaches having shown strong performance in classification of standalone objects, it is unclear how to extend them to work reliably in larger scenes (e.g. with covered objects) and on per-point tasks such as part segmentation [23].

A more recent approach, pioneered by PointNet [23], is feeding raw point cloud data into neural networks. As point clouds are unordered sets, these networks have to be permutation invariant - PointNet achieves this by using the max-pooling operation to form a single feature vector representing the global context from a variable amount of points. PointNet++ [24] proposes an extension that introduces local context by stacking multiple PointNet layers. Further improvements were made by introducing Dynamic Graph CNNs (DGCNNs) [33], in which a graph convolution is applied to edges of the k-nearest neighbor graph of the point clouds, which is dynamically recomputed in feature space after each layer. Similar performance was achieved by PointCNN [16], which uses a hierarchical convolution that is trained to learn permutation invariance. All neural networks operating on raw point cloud data naturally provide per-point embeddings, making them particularly useful for point segmentation tasks. Our proposed method can leverage these methods as it is flexible with regards to the use of specific neural network architecture.

### 2.2  Unsupervised and Self-Supervised Deep Learning

Deep learning algorithms have demonstrated the ability to learn powerful internal hierarchical embeddings through unsupervised learning tasks, in which no supervision is given at all, or self-supervised tasks, where the labels are generated from the data itself [14, 7, 9]. These representations can be directly used in downstream tasks or as strong initializers for supervised tasks [20, 8]. In cases where large amounts of data are available but annotated samples are scarce, unsupervised or self-supervised learning can significantly reduce the number of annotated training samples required for strong performance in various tasks [37], making such methods particularly desirable for point clouds.

Following the impressive results that have been achieved with GANs [10] and Autoencoders [13] in the image domain, previous efforts for unsupervised learning on point clouds have been adaptations of these approaches. However, GANs for point clouds have been limited to either work on voxelized representations [34], on 2D-rendered images of point clouds [12], or through adversarial learning on the learned embedding space from an external Autoencoder [1] as sampling unordered but intra-dependent sets of points with neural networks is non-trivial. Autoencoders on the other hand work by learning to encode inputs into a latent space before reconstructing them, therefore requiring similarity or reconstruction metrics. Besides Autoencoders on voxelized representations [29] in which conventional loss functions can be applied per-voxel, Autoencoders have also been applied on raw point clouds [37, 15]. When operating on raw point clouds, Autoencoder-based methods for point clouds rely on similarity metrics such as the Chamfer (pseudo) distance, which acts as a differentiable approximation to the computationally infeasible Earth Mover's Distance [26]. Computing the Chamfer distance can be limited by memory requirements in large point clouds, but more importantly, the authors [1] observe that specific pathological cases are handled incorrectly. This motivates self-supervised methods such as ours which avoid potentially problematic similarity functions.

A completely different approach to self-supervised learning in the image domain was taken by [7], in which a neural network is trained to predict the spatial relation between two randomly chosen image patches. The authors demonstrate the effectiveness of the learned features in a range of experiments and argue that such a classification task tackles the problem of the extremely large variety of pixels that can arise from the same semantic object in images. This holds even more true when moving from images to point clouds, i.e. from regular grids in 2D space to unordered sets in 3D space. These ideas were extended in [21], where a neural network with a limited receptive field was trained to correctly place randomly displaced image patches to their original position. The authors of [7, 21] identify the challenge of trivial solutions for such self-supervised tasks in the image domain, such as chromatic aberration or the matching of low-level feature such as the position of lines in image segments. They take extensive precautions to alleviate this problem, one of which is limiting the receptive field of the neural network, which prevents the same neural network used for pre-training from being used without any changes in further supervised training. Another approach for self-supervised learning was taken by [9], in which a neural network learns to identify the correct rotation on an image. However, this approach is limited to domains in which a clear height-axis is defined. We build on the concepts of [21] and adapt the idea of reordering patches to point clouds, which have certain characteristics that make them particularly well-suited for such a task.

## 3  Method

---
**Algorithm 1:** Generation of Self-Supervised Labels

---
1: **function** GET_SELF_SUPERVISED_LABEL($X \subset \mathbb{R}^3, k \in \mathbb{R}$)
2:     $X_1 \leftarrow$ scale_to_unit_cube($X$)
3:     $X_1, y \leftarrow$ voxelize($X_1, k$)                    ▷ get corresponding voxel ID for each point in $X$
4:     $\pi \leftarrow$ random_permutation($0..k^3$)
5:     **for** i in $0..k^3$ **do**
6:        new_position $\leftarrow$ move_to_voxel($X_1[i], \pi[y[i]])$)
7:        $X_1[i] \leftarrow$ augment(new_position)
8:     **return** $X_1, y$

---

In this paper we propose a self-supervised method that learns powerful representations from raw point cloud data. Our method works by training a neural network to reassemble point clouds whose parts have been randomly displaced. The key assumption of the proposed method is that learning to reassemble displaced point cloud segments is only possible by learning holistic representations that capture the high-level semantics of the objects in the point cloud.

We phrase the self-supervised learning task as a point segmentation task, in which the label for each point is generated from the point cloud itself with the following procedure: the input point cloud is scaled to unit cube before each axis is split into $k$ equal lengths, forming $k^3$ voxels. We use these to assign each point its voxel ID as a label. Subsequently all voxels are randomly swapped with other voxels and a neural network is trained to predict the original voxel ID of each point. The points in each voxel can also be augmented (e.g. randomly shifted by a small amount) to improve generalization. Pseudo-code for this entire procedure is provided in Algorithm 1. Note that using the voxel ID as per-point label admits a unique solution even for almost all axis-symmetric point clouds, as long as the individual voxels are not all randomly rotated, i.e. as long as a general sense of the orientation of the input point cloud is maintained. While $k$ may be varied across domains, depending on the amount of detail in the input point clouds, we list all results with $k = 3$. Additional details are discussed in Section 5.

The proposed method is agnostic of the specific neural network architecture at hand - any neural network capable of point segmentation tasks, such as PointNet [23], PointNet++ [24], DGCNN [33], or PointCNN [16] can be used out-of-the-box. These network architectures can be pre-trained in a self-supervised manner with our method and used as-is for further supervised training. Furthermore, as point clouds do not suffer from the same trivial solutions as identified in the image domain by [7, 21], no limitation is needed on the receptive field size. Phrasing the self-supervised task as a point segmentation task brings many advantages: there is no reliance on possibly flawed similarity metrics as with Autoencoders, it is not necessary to sample unordered sets of points from a neural network as with GANs, and the method can work on raw point cloud data and does not require voxelized

Table 1: Comparison of our method against previous unsupervised methods in downstream object classification on the ModelNet40 and ModelNet10 dataset in terms of accuracy. A linear SVM is trained on the representations learned in an unsupervised manner on the ShapeNet dataset.

| Model | MN40 | MN10 |
|---|---|---|
| VConv-DAE [29] | 75.50% | 80.50% |
| 3D-GAN [34] | 83.30% | 91.00% |
| Latent-GAN [1] | 85.70% | **95.30%** |
| FoldingNet [37] | 88.40% | 94.40% |
| VIP-GAN [12] | 90.19% | 92.18% |
| PointNet + Pre-Training (Ours) | 87.31% | 91.61% |
| DGCNN + Pre-Training (Ours) | **90.64%** | 94.52% |

or 2D-rendered representations of point cloud, making our approach universally applicable to any point cloud data. Operating on raw point cloud enables flexibility with regards to the point cloud density and allows for learning of per-point embeddings instead of per-voxel or per-pixel embeddings without explicit supervision.

## 4 Experiments

### 4.1 Object Classification

In this section, we show that the embeddings learned with our method outperform state-of-the-art unsupervised methods in a downstream object classification task and demonstrate the benefits of pre-training with our method before fully supervised training. In line with previous approaches, we evaluate our performance on the object classification problem using the ModelNet dataset [35], which contains CAD models from different categories of man-made objects. For this we use the standard train/test split, with the same uniform point sample as defined in [23] with ModelNet40 on 40 classes containing 9843 train and 2468 test models and ModelNet10 on ten classes containing 3991 and 909 models respectively.

In the first experiment, we follow the same procedure as in [1, 34, 37, 12]. We train a model in a self-supervised manner on the ShapeNet dataset [5], which consists of 57448 models from 55 categories. After that, we train a linear Support Vector Machine (SVM) [6] on the obtained embeddings of the ModelNet40 train split and evaluate it on the test split. We do this with a PointNet and a DGCNN with the exact same setup as proposed by the authors for object classification [33, 23], the object embeddings are obtained after the last max-pooling layer. This experiment evaluates the learned embeddings in a transfer learning task, demonstrating their generalizability. From every model in ShapeNet we use the same random sample of 2048 points on the model surface as provided by [37]. The results are displayed in Table 1. Our method outperforms all previous approaches on ModelNet40, and all except Latent-GAN on ModelNet10. However, as noted by [37], the point cloud

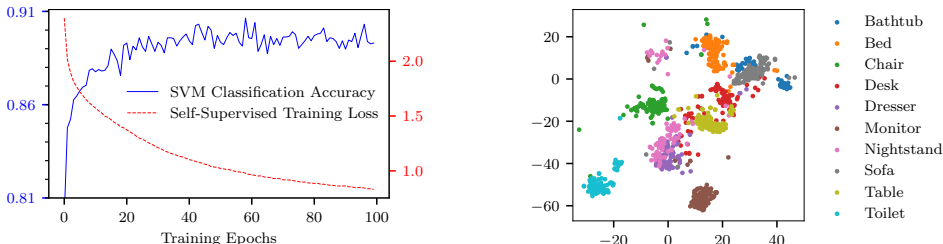

(a) The self-supervised training loss on the ShapeNet dataset and the linear SVM accuracy trained on obtained embeddings for the ModelNet dataset. Performing better on the unsupervised tasks results in stronger embeddings for downstream object classification.

(b) Visualization of the object embeddings of the ModelNet10 test data obtained through training with the proposed self-supervised method on the ShapeNet dataset. t-SNE with perplexity 10 and 1000 iterations was used for dimensionality reduction.

Figure 2

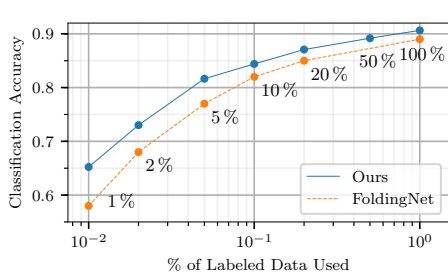
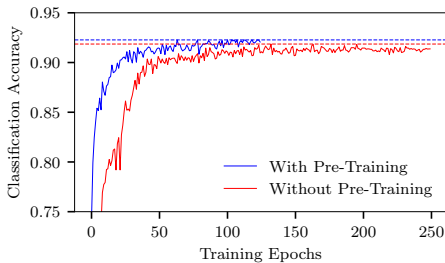

(a) Figure showing how the linear SVM classification accuracy for ModelNet40 behaves when few annotated training samples are available.

(b) The training curves on the ModelNet40 object classification task of a DGCNN pre-trained with our self-supervised method (blue) on the ShapeNet dataset and a randomly initialized DGCNN (red).

Figure 3

format and sampling procedure from Latent-GAN is not publicly available, making a comparison on ModelNet10 accuracy inconclusive. Figure 2a shows that a decrease in self-supervised training loss on ShapeNet gives a better downstream classification accuracy on ModelNet40, which suggests that correctly reconstructing the point cloud parts results requires learning representations that capture the semantics of the objects at hand. The obtained embeddings from a DGCNN with out method for the ModelNet10 test data are visualized using t-SNE [17] in Figure 2b. One can see that clear, separable clusters are formed for each class except for the classes dresser (violet) vs nightstand (pink), which are almost visually indiscernible when scaled to unit cube, as done in the ShapeNet dataset.

In a second experiment, we show that a very small number of labelled samples can suffice to achieve strong performance in a downstream task, which is one of the main motivations of self-supervised learning. We evaluate our method in such a scenario by limiting the number of training samples available in the ModelNet object classification task. We sample according to the following procedure: first we randomly sample one object per class, and then sample the remaining objects uniformly out of the entire training set. We compare the performance of a linear SVM trained on the embeddings obtained from training a DGCNN on ShapeNet with our method to those obtained with FoldingNet [37] in Figure 3a. The embeddings obtained from our method lead to higher accuracy than those obtained with FoldingNet with any amount of training labels. Using only 1 % of training data, equivalent to three or less samples per class, our model is able to achieve 65.2 % accuracy on the test set. When using 10 % of available training samples, this accuracy rises up to 84.4 %.

Finally, we demonstrate the benefit of pre-training with our method, by pre-training a DGCNN in a self-supervised manner on the ShapeNet dataset with 1024 points chosen randomly from each model for 100 epochs before fully supervised training on the ModelNet40 dataset. As seen in 3b, self-supervised pre-training acts as a strong initializer, reducing the number of supervised epochs needed for strong performance and even improving the final object classification accuracy with DGCNN (Table 2).

## 4.2 Part Segmentation

In this section we explore the per-point embeddings obtained through unsupervised training in a part segmentation task. Again, we train our model in a self-supervised fashion on the ShapeNet

Table 2: Comparison to state-of-the-art supervised methods in ModelNet40 classification accuracy. All models are trained and evaluated on 1024 points. Self-supervised pre-training is performed on the ShapeNet dataset.

| Model | Accuracy |
|---|---|
| PointNet [23] | 89.2% |
| PointNet++ [24] | 90.7% |
| PointCNN [16] | 92.2% |
| DGCNN + Random Init [33] | 92.2% |
| DGCNN + Pre-Training (Ours) | **92.4**% |

Table 3: The effect of pre-training on ShapeNet Part Segmentation. Metric is mean IoU% of parts per object class.

| | Mean | Aero | Bag | Cap | Car | Chair | Earphone | Guitar | Knife | Lamp | Laptop | Motor | Mug | Pistol | Rocket | Skateboard | Table |
|---|---|---|---|---|---|---|---|---|---|---|---|---|---|---|---|---|---|
| # Shapes | | 2690 | 76 | 55 | 898 | 3758 | 69 | 787 | 392 | 1547 | 451 | 202 | 184 | 283 | 66 | 152 | 5271 |
| PointNet | 83.7 | 83.4 | 78.7 | 82.5 | 74.9 | 89.6 | 73.0 | **91.5** | 85.9 | 80.8 | 95.3 | 65.2 | 93.0 | 81.2 | 57.9 | 72.8 | 80.6 |
| PointNet++ | 85.1 | 82.4 | 79.0 | 87.7 | **77.3** | 90.8 | 71.8 | 91.0 | 85.9 | **83.7** | 95.3 | **71.6** | **94.1** | 81.3 | 58.7 | 76.4 | **82.6** |
| DGCNN | 85.1 | **84.2** | 83.7 | 84.4 | 77.1 | **90.9** | 78.5 | **91.5** | **87.3** | 82.9 | **96.0** | 67.8 | 93.3 | **82.6** | 59.7 | 75.5 | 82.0 |
| Ours | **85.3** | 84.1 | **84.0** | **85.8** | 77.0 | **90.9** | 80.0 | **91.5** | 87.0 | 83.2 | 95.8 | **71.6** | 94.0 | **82.6** | **60.0** | **77.9** | 81.8 |

dataset. The supervised task is then to correctly classify each point of an object into the correct object part on the ShapeNet Part dataset [38], which is a subset of the full ShapeNet containing 16881 3D objects from 16 categories, annotated with 50 parts in total. We use the official train / validation / test splits [38]. Following the same procedure as in [23, 24, 33], the one-hot encoded object class label of the object is given as an input during supervised training. During the 200 epochs of pre-training, a random class label is given to each object. Part segmentation is evaluated on the mean Intersection-over-Union (mIoU) metric, calculated by averaging IoUs for each part in an object before averaging the obtained values for each object class. The results are shown in Table 3. A DGCNN pre-trained with our method slightly outperforms a randomly initialized DGCNN, the differences in accuracy being particularly notable on the classes with few samples.

In Figure 4 we show a visualization of the features learned for objects after self-supervised training but before any fully supervised training. The visualizations are obtained by selecting a random point and visualizing the distance to the two (sequentially chosen) furthest points in the learned feature space using a color scale. The visualizations show that the features learned in a self-supervised manner can capture high-level semantics such as object parts without ever having seen part IDs. In Figure 5 a visualization of the features for each point from ten airplanes and ten chairs is shown. The features are projected into two dimensions using UMAP [19]. One can clearly see that the two object classes form clear, separable clusters in the feature spaces and that clear, discernible clusters are formed for the individual object parts. Individual objects from the classes are not identifiable, showing that the learned features generalize over reoccurring structures. This highlights the semantics of the high-level features learned with our method.

## 4.3   Semantic Segmentation

In this semantic segmentation task we evaluate the effectiveness on our method on data that goes beyond simple, free-standing objects. The task is evaluated on the Stanford Large-Scale 3D Indoor Spaces (S3DIS) dataset [2]. The dataset consists of 3D point cloud scans from 6 indoor areas totalling 272 rooms. The points are classified into 13 semantic classes such as board, chair, ceiling, beam, and clutter. Each room is split into blocks of $1m \times 1m$ area and each point is given as a 6D vector containing XYZ coordinates and RGB color values. In this setup we evaluate the case in which there is large amounts of unlabelled data and only few annotated samples are available. For this the largest area (area 5) is chosen as the test set, and the other areas form distinct training sets. We compare two

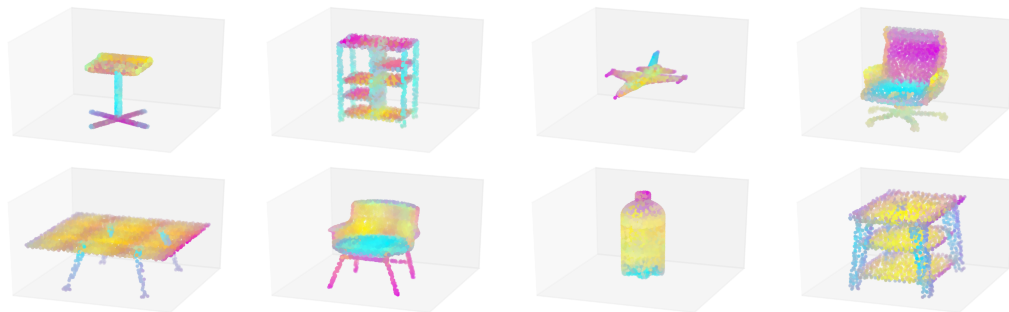

Figure 4: A visualization of the features learned through self-supervised training with our method for individual objects. A color scale shows the distance in feature space between a randomly sampled point and its two (mutually) furthest neighbors in feature space.

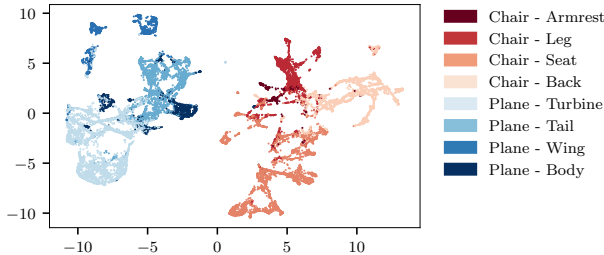

Figure 5: Visualization of the per-point features of 10 airplanes and 10 chairs from the ShapeNet Part dataset. UMAP is used for dimensionality reduction for visualization purposes.

Table 4: Results of semantic segmentation on the S3DIS dataset. Results are evaluated on area 6.

| Supervised Train Area | # Samples | Random Init | | Pre-Training (ours) | |
|---|---|---|---|---|---|
| | | mIoU% | Acc % | mIoU% | Acc % |
| Area 1 | 3687 | 43.6% | 82.9% | **44.7%** | **83.5%** |
| Area 2 | 4440 | 34.6% | **81.2%** | **34.9%** | **81.2%** |
| Area 3 | 1650 | 39.9% | 82.8% | **42.4%** | **84.0%** |
| Area 4 | 3662 | 39.4% | 82.8% | **39.9%** | **82.9%** |
| Area 6 | 3294 | **43.9%** | 83.1% | **43.9%** | **83.3%** |

DGCNNs with the architecture proposed for semantic segmentation by the authors for each training area, one that has been pre-trained on all areas except area 5, and one that is not pre-trained. The task is evaluated in mIoU% per object class and total per-point classification accuracy. The results are shown in Table 4. Pre-training improves the mIoU and classification accuracy in all cases except two, in which the two methods are tied. As expected, the difference is the largest for area 3, where the number of training samples for fully supervised learning is the smallest.

## 5   Discussion

Throughout all experiments, our proposed method learns representations that prove to be effective. This leads us to believe that trivial solutions to the task of reconstructing the inputs, as discussed for the image domain by [7, 21] are not a significant problem for point clouds. Point clouds do not suffer from chromatic aberration and point cloud parts can be shifted and rotated freely in the coordinates, alleviating the issue of simply matching lines and edges. In this paper we performed all experiments with a three-by-three voxel grid during self-supervised pre-training, which we observed to outperform both $k = 2$ and $k = 4$. We found that randomly rotating 15% of the individual voxels and randomly replacing one voxel in each input point cloud with a random voxel from a randomly drawn input point cloud from the same dataset leads to a slightly higher quality of the embeddings in the object classification task (consistently around 0.2% SVM accuracy in the downstream object classification task), therefore we kept this setup throughout all experiments. An extensive evaluation on how to fine-tune the self-supervised task to a specific dataset or domain is not the focus of this paper, instead we show that our simple approach works reliably in all evaluated cases.

## 6   Conclusion

In this paper we propose a self-supervised method for learning representations from unlabelled raw point cloud data. In this easy-to-implement method, a neural network learns to reconstruct input point clouds whose parts have been randomly displaced. While solving this task, high-level representations of the underlying input point clouds are learned. We demonstrate the effectiveness of the learned representations in downstream tasks and show our method can improve the sample efficiency and the accuracy of state-of-the-art models when used to pre-train with large amounts of data before fully supervised training. As our method is independent of the specific neural network architecture, we expect to see further benefits of using our results as more effective neural networks for processing raw point cloud data are developed in the future.

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
