[Reviews · NeurIPS 2019]

Reviewer 1



Originality: this work is an extension of "Unsupervised learning of visual representations by solving jigsaw puzzles" in 3D point cloud. The design is pretty natural and the paper has shown 3D design could be even simpler than its 2D counterpart since trivial solutions will not easily appear. The paper also shows through various experiments that the learned feature could be very helpful for different downstream tasks and is quite general for different learning frameworks. The overall contribution is valid yet a bit incremental. The references are ok as far as I could tell. Quality: the paper is technically sound with enough technical details provided. The paper requires prealigned Clarity: the paper is clearly written and can be easily followed. Significance: self-supervised learning in 3D point cloud is an important problem in 3D computer vision. This paper has adapted an existing approach from 2D image world to process 3D point cloud. The approach is quite simple and natural. The change from 2D images seems quite small and the induced improvement is a bit marginal as well. The experiments are quite thorough though covering different 3D processing tasks and various baselines. I would say the paper has made a valid yet quite limited improvement without too many inspiring points, which would limit its influence in the community.

Reviewer 2



Originality: This paper is a novel combination of an existing method [7,21] for 2D images, to an existing task (point cloud feature learning). Given the success of [21], one would expect it also works for 3D representation where the spatial layout is equally or more important, which is confirmed by the results in this paper. The citations in this paper sufficiently cover related work. Quality: Most of the experimental results appear to be meaningful and support claimed advantages of this method: architecture-agnostic, avoids reconstruction metric, helps supervised down-stream tasks. But the comparison to alternative methods in Table 1 is weakened by the fact that model architectures used by the baseline methods are not mentioned. Given the significant gap between PointNet + Pre-training vs DGCNN + Pre-Training, I wonder how much of the improvement simply comes from a better architecture (DGCNN). For example, FoldingNet which uses a PointNet-like architecture is actually better than this method + PointNet as shown in Table 1. There's also a minor problem: page 4 L 152 "no limitation is needed on the receptive field size" is not supported by any analysis/results, it would be helpful to mention the receptive field size of the two base architectures studied in this paper. Clarity: The paper is well organized and has provided enough details for reproducing this results. Significance: This paper is addressing a important problem that has potentially big impact. However I'm not confident if it is advancing the state-of-the-art due to a concern stated above. Update after rebuttal: Thanks for providing the numbers I requested in the original review. Changed my overall scores accordingly.

Reviewer 3



This paper proposed a self-supervised method for learning representation from unlabelled point clouds. By random displacing the point clouds and training a network to reconstruct them, good feature representation can be learned, which can benefit the downstream tasks. Self-supervised learning is a hot topic in recent years. The authors extended the idea of [21] to 3D point clouds and showed its effectiveness. The proposed method is simple and easy-to-implement. The main weakness is that the performance gain is limited according to table II and III.

[Author Response · NeurIPS 2019]

We thank the reviewers for their constructive reviews which clearly show that all reviewers have thoroughly read the paper and are very familiar with related work. In our comments we address the reviews in the order of the reviewer number.

**Reviewer #1**

The reviewer notes that it would be great to include an analysis about how the pre-alignment of the objects in the ModelNet [35] and ShapeNet [5] dataset influences the quality of the learned representations. In previous literature, supervised tasks have consistently benefitted from pre-aligned objects by a small but significant margin (e.g. 0.3% classification accuracy on ModelNet40 in [16]) when compared to objects with random orientation. This very small gap was corroborated in our initial experiments, we therefore stuck to reporting pre-aligned objects in the paper. We are currently re-running the experiments with random orientation and will provide the numbers in the camera-ready version.

The reviewer also states that our paper could benefit from including PointNet++ [24] in the result tables to further highlight the architecture-agnosticity. We will include these results in the camera-ready version, as the required time for training PointNet++ for all provided experiments exceed the length of the author response period.

**Reviewer #2**

The reviewer notes that on page 4 line 152, the phrase "no limitation is needed on the receptive field size" would benefit from additional clarification in the context of our paper. In fact, the DGCNN and PointNet architectures used in our comparison use a max-pooling layer over all points in the input point cloud. The receptive field of the used architectures is therefore the entire input point cloud. In contrast, in [21] within the context of images, the receptive field is limited such that the neural network can only use the information from a single image patch. We will explicitly mention this in the camera-ready version.

The reviewer points out that there is a significant performance gap between using our method with PointNet and with DGCNN. The reviewer wonders how much of the performance gain of our method with regards to previous unsupervised methods (i.e. FoldingNet) really stems from the improved architecture of DGCNN or the proposed task. The reviewer argues that FoldingNet can outperform our method when used with PointNet even though it has a PointNet-like architecture, however we believe that FoldingNet has an architecture that can be better compared to DGCNN than to PointNet as it also uses graph convolutions. The FoldingNet decoder alone has 1.05M parameters. Unfortunately the parameters count of the encoder are not stated, but the code indicates around 0.65M parameters which results in 1.7M parameters in total whereas DGCNN only has 1.55M - nonetheless our method outperforms FoldingNet by a significant margin. For some previous unsupervised methods (e.g. VIP-GAN [12]), no detailed description of the architecture and no code is provided. We will do our best to provide the number of parameters and layers for each of the previous unsupervised methods in the camera-ready version by contacting the authors of these papers or re-implementing them to the best of our ability.

**Reviewer #3**

The reviewer points out that some additional ablation study might be beneficial. In terms of dependence on neural network architectures and training procedures, we did not modify those proposed in PointNet [23] and DGCNN [33]. With the inclusion of PointNet++ [24] in the camera-ready version, we will further highlight that the proposed method is architecture-agnostic. The reviewer also mentions that combining the proposed method with other ideas to further improve the performance of the neural network could be beneficial. We leave this as future work, as combining self-supervised tasks into a multi-task context is out of scope for this paper, whose main goal is instead to show in isolation that the proposed self-supervised learning task can be used flexibly in the context of deep learning on point clouds without particular fine-tuning to tasks or data-domains.

[Meta-Review · NeurIPS 2019]

The paper received weak but still positive support from reviewers. The main concern was limited novelty on transferring work into 2D computer vision to 3D computer vision. However, the simplicity of the approach is a strength, and the approach seems to work well, which the reviewers generally agree on.